# Biological Control as Part of the Soybean Integrated Pest Management (IPM): Potential and Challenges

Adeney de F. Bueno [1,*], Weidson P. Sutil [2], Simone M. Jahnke [3], Geraldo A. Carvalho [4], Maria Fernanda Cingolani [5], Yelitza C. Colmenarez [6] and Natália Corniani [6]

1 Empresa Brasileira de Pesquisa Agropecuária—Embrapa Soja, Londrina 86085-981, PR, Brazil
2 Programa de Pós-Graduação em Entomologia, Universidade Federal do Paraná (UFPR), Curitiba 91531-980, PR, Brazil; plauter80@gmail.com
3 Departamento de Fitossanidade, Faculdade de Agronomia, Universidade Federal do Rio Grande do Sul, Porto Alegre 915400-000, RS, Brazil; mundstock.jahnke@ufrgs.br
4 Departamento de Entomologia, Universidade Federal de Lavras, Lavras 37203-202, MG, Brazil; gacarval@ufla.br
5 Centro de Estudios Parasitológicos y de Vectores (CEPAVE), La Plata B1900, Buenos Aires, Argentina; fernandacingolani@cepave.edu.ar
6 CABI Latin America, FEPAF—Avenida Universitária, Botucatu 18610-307, SP, Brazil; y.colmenarez@cabi.org (Y.C.C.); n.corniani@cabi.org (N.C.)
* Correspondence: adeney.bueno@embrapa.br; Tel.: +55-43-3371-6208

**Abstract:** Soybean production is usually performed on large scales, requiring simple but efficient pest management to be successful. Soybean fields are inhabited by several species of arthropods, demanding constant development of management practices to prevent pest outbreaks. More recently, stink bugs have become the most important pest group of soybeans in the Neotropics, responsible for up to 60% of the applied insecticides in Brazil. Natural enemies represent an important mortality factor that can keep the damage caused by stink bugs below the economic threshold levels without additional control actions. Thus, Conservation Biological Control (CBC) strategies can be adopted to preserve or even promote the increase in such natural enemies in the fields, or alternatively, massive releases of biocontrol agents in Augmentative Biological Control (ABC) programs could be adopted. Simple practices such as reducing insecticide use (with the adoption of economic thresholds), prioritizing harmless insecticides or biopesticides, and planting resistant soybean cultivars have been adopted in Brazil with positive results. The challenges to increasing the adoption of more complex stink bug management in commodity crops such as soybean may be overcome using the more recent economic incentives in the global agenda of decarbonized agriculture. The potential and challenges of conservation and augmentative biological control are further discussed in this review.

**Keywords:** hymenoptera; Scelionidae; sustainability; preservation; insecticide mitigation; insecticide selectivity; economic thresholds

## 1. Introduction

Until recently, chemical control has been farmers' first line of defense to control pests in the production of soybean (*Glycine max* (L.) Merrill) [1,2], the most important source of plant-based protein and vegetable oil worldwide [3]. The stink bug species of the genus *Euschistus* and *Diceraeus* (Hemiptera: Pentatomidae) [2] are major pests that damage soybeans and require a greater amount of insecticides in South America, mainly in the central region of Brazil at latitudes between 0° and 23° [4] and in the Southeastern USA [5]. In addition to *Euschistus* sp. and *Diceraeus* sp., at least 54 other stink bug species from different genus have been reported attacking soybeans [6]. Feeding directly from soybean pods, those insects can significantly impact the crop, reducing yields by 15% [7] and impairing the physiological and sanitary quality of the seeds when not properly managed [8].

Despite the high potential of damage, insecticides used in an uncoordinated manner and without considering the mechanisms of action can impact human health and the environment [9,10]. Moreover, the overuse of insecticides can reduce natural biocontrol agents [11] and pollinators [12], induce pest resistance [13], and trigger pest resurgence and/or outbreaks of secondary pest species, in addition to other negative side effects [2]. Consequently, reducing synthetic chemical use in agriculture to produce food inexpensively and sustainably [14] has become a global goal [15]. Thus, more sustainable stink bug management tools are of high theoretical and practical interest and will benefit thousands of soybean farmers. Biological control is the most adopted sustainable pest control strategy available for agricultural use [16].

The three commonly adopted ways to apply (or exploit) biological control in the agroecosystem are (1) conservation biological control (CBC), which consists of preserving or favoring the existing natural enemies; (2) classical biological control, by introducing new natural enemies to establish a permanent population into the new agroecosystem; and (3) augmentative biological control (ABC), which relies on massive and periodic releases of biocontrol agents to rapidly reduce pest population [17]. ABC is drastically increasing worldwide, and the global ABC market is expected to surpass USD 10 billion in 2027 [18]. However, the successful adoption of ABC programs depends on CBC practices, which provide more balanced and equilibrated agroecosystems [2]. The establishment of more resilient agroecosystems requires actions that consider the system as a whole and improve agriculture through what is known as Landscape Architecture (LA) [19]. Originally, the idea of LA represented a form of agriculture that applied science-based agricultural practices while considering the aesthetic dimension of rural landscapes in relation to the necessary food production [20]. Subsequently, the concept of integrated landscape management emerged, combining food production with the conservation of ecosystem services, particularly those supplied by habitat biodiversity [21].

Among LA procedures, CBC is highlighted for its beneficial impact on agroecosystems from both ecological and economic standpoints [2,11]. CBC is based on preserving and improving natural biological control by maintaining a habitat that can sustain natural enemies [22]. The concept of CBC neither necessarily excludes other types of pest management nor the introduction of other biological control agents via ABC strategies [23]. Instead, CBC helps to provide a more favorable environment for biocontrol agents to survive and prosper, contributing to more effective ABC practices [24]. A variety of management practices are needed to manipulate the agricultural habitat in favor of biocontrol agents of pests in the agroecosystem to enhance their fitness and optimize their impact on pest population growth [25]. Despite negative interactions that can exist among different species of natural enemies [26], ABC and CBC strategies are more likely to be compatible with each other than chemical control [11]. Although, in some cases, multiple field experiments have shown negative impacts of chemical insecticides on biological control [27], some selective insecticides have minimal impact on biocontrol agents due to their physiological selectivity. Furthermore, non-selective insecticides can be sprayed in a way that reduces contact with beneficial organisms and, consequently, has a lower impact on them, i.e., a strategy called ecological selectivity [11].

Stink bugs are considered hard-to-kill insects and are the target of up to 60% of all insecticide applications performed on soybean fields in areas where they occur [28]. It is of great theoretical and practical interest to discuss sustainable alternatives to improve their management. Biological control is of crucial importance to agroecosystem sustainability. Some CBC and ABC practices can be adopted within integrated pest management (IPM) to enhance the impact of biological control in soybeans. Therefore, the following discussion is based on a systematic review of published articles and practical knowledge regarding the importance and challenges of adopting biocontrol and preserving natural enemies among the soybean IPM strategies for successful sustainable management of stink bugs.

## 2. The Importance of Biocontrol Agents of Stink Bugs in Soybean Production

Several organisms, such as fungi, viruses, bacteria, and arthropods, can act as biocontrol agents of soybean pests, infecting, preying on, or parasitizing pests in their different developmental stages [17]. The capacity of those beneficial organisms to reduce pest population, known as biological control, usually plays an important role in limiting the densities of pests in agriculture. Biocontrol is usually safer for the environment and more specific against the target species than chemicals [11]. Thus, knowledge about the species of biocontrol agents and their specific potential to reduce the population of each economically important stink bug species is critical for the correct adoption of CBC or ABC strategies, aiming at the benefits from the potential of those biocontrol agents in sustainably reducing stink bug population in soybean fields [17].

A large and diverse complex of natural enemies and stink bug pests has been recorded in the most important soybean-producing countries (Brazil and the USA). Stink bug eggs, late nymphal stages, and adults are most commonly attacked by parasitoids (Table 1), whereas predators prefer eggs and early nymph stages [5].

**Table 1.** Stink bug parasitoids recorded in Brazil or the USA.

| Species | [1] Host Species | [2] Stage(s) Attacked | Parasitism Rate | Country | Reference |
|---|---|---|---|---|---|
| *Anastatus mirabilis* (Walsh & Riley) (Hymenoptera: Eupelmidae) | *E* | E | 0.8% | USA | [29] |
| *Anastatus reduvii* (Howard) (Hymenoptera: Eupelmidae) | *Ch* | E | 44% | USA | [30] |
| *Aridelus rufotestaceus* Tobias (Hymenoptera: Braconidae) | *E; Nv* | N | No information | USA | [31,32] |
| *Gryon obesum* Masner (Hymenoptera: Scelionidae) | *E* | E | 1.4% to 2.4% | USA | [33] |
| | *E* | E | 2.6% | USA | [29] |
| *Hexacladia hilaris* Burks (Hymenoptera: Encyrtidae) | *Ch; Nv* | N; A | No information | Brazil and USA | [34–37] |
| *Hexacladia smithii* Ashmead (Hymenoptera: Encyrtidae) | *E; Eh* | A | 0.6% to 90% | Brazil and USA | [34–36,38–41] |
| | *Em* | A | No information | Brazil | [4] |
| *Ooencyrtus anasae* (Ashmead) (Hymenoptera: Encyrtidae) | *Nv; Pz* | E | No information | Brazil | [42] |
| *Ooencyrtus submetalicus* (Howard) (Hymenoptera: Encyrtidae) | *Nv* | E | No information | USA | [34] |
| | *Em* | E | No information | Brazil | [43] |
| *Telenomus edessae* Brèthes (Hymenoptera: Scelionidae) | *Em* | E | 0.4% | Brazil | [39] |
| *Telenomus podisi* Ashmead (Hymenoptera: Scelionidae) | *Nv* | E | No information | USA | [34] |
| | *Nv* | E | 11.5% to 100% | USA | [33] |
| | *Ch* | E | 3.4% | Brazil | [39] |
| | *Dm* | E | 25% to 50% | Brazil and USA | [39,44] |
| | *Eh* | E | 43.4% | Brazil | [39] |
| | *Eh* | E | 59.3% to 62.5% | Brazil | [45] |
| | *E* | E | 77.8% | USA | [46] |
| | *E* | E | 69% to 100% | USA | [33] |
| | *Pz* | E | 20.9% | Brazil and USA | [39,47] |
| | *Pz* | E | 23.8% to 39.5% | Brazil | [45] |
| | *Pz* | E | No information | Brazil | [42] |

**Table 1.** *Cont.*

| Species | [1] Host Species | [2] Stage(s) Attacked | Parasitism Rate | Country | Reference |
|---|---|---|---|---|---|
| *Trissolcus basalis* (Wollaston) (Hymenoptera: Scelionidae) | *Nv* | E | 74.5 | USA | [46] |
| | *Nv* | E | 25% to 100% | USA | [33] |
| | *Nv* | E | 53.8% | | |
| | *Dm* | E | 16.7% | | |
| | *Eh* | E | 10.6% | Brazil and USA | [39,47] |
| | *Ch* | E | 24% | | |
| | *Pz* | E | 22.8% | | |
| | *Th* | E | 23.1% | | |
| | *E* | E | 3% to 100% | USA | [33] |
| | *E* | E | 18.6% | USA | [46] |
| *Trissolcus brochymenae* (Ashmead) (Hymenoptera: Scelionidae) | *E* | E | 7.4% | USA | [48] |
| *Trissolcus edessae* Fouts (Hymenoptera: Scelionidae) | *E* | E | 19.3% | USA | [29] |
| | *E* | E | 6.6% | USA | [46] |
| | *E* | E | 3.1% | USA | [29] |
| | *Ch* | E | 35% | USA | [30] |
| *Trissolcus elimatus* (Johnson) (Hymenoptera: Scelionidae) | *Em* | E | No information | Brazil | [43] |
| *Trissolcus euschisti* (Ashmead) (Hymenoptera: Scelionidae) | *E* | E | 20% | USA | [46] |
| | *E* | E | 3.4% | USA | [48] |
| | *E* | | 5.3% to 20% | USA | [33] |
| | *E* | E | 3.6% | USA | [29] |
| | *Em* | E | No information | Brazil | [43] |
| *Trissolcus teretis* Johnson (Hymenoptera: Scelinonidae) | *Pz* | E | No information | Brazil | [42] |
| *Trissolcus thyantae* Ashmead (Hymenoptera: Scelionidae) | *E* | E | 3.4% | USA | [48] |
| | *E* | E | 1.1% to 5.9% | USA | [48] |
| | *E* | E | 5.9% | USA | [29] |
| | *Nv* | E | 1.4% to 8.3% | USA | [33] |
| *Trissolcus urichi* (Crawford) (Hymenoptera: Scelionidae) | *Em* | E | 14.2% | Brazil | [39] |
| | *Em* | E | No information | Brazil | [43] |
| | *Nv; Eh; Pz* | E | No information | Brazil and USA | [42,47] |
| | *Eh* | E | No information | Brazil | [49] |

[1] Stink bug species abbreviations: *Ch*: *Chinavia* sp., *Dm*: *Diceraeus melacanthus*, *Em*: *Edessa meditabunda*, *E*: *Euschistus* sp., *Eh*: *Euschistus heros*, *Nv*: *Nezara viridula*, *Pz*: *Piezodorus guildinii*, *Th*: *Thyanta* sp. [2] E: egg, N: nymph, and A: adult.

Many microhymenopteran species are parasitoids of stink bugs and are mostly egg parasitoids. Twenty-three species of stink bug egg parasitoids have already been identified in soybeans [50], and these natural enemies are considered the most important biocontrol agents for this group of pests [17,51,52]. They are often responsible for naturally keeping stink bug populations below economic injury levels (without needing to adopt any additional control strategy) [50].

In the Neotropical region, *Nezara viridula* (Linnaeus, 1758) (Pentatomidae) was the most abundant species until 1999, accounting for 44% of the stink bugs recorded in soybean areas [53]. In the surveys performed in that era, *T. basalis* was the most important egg parasitoid species, responsible for more than 90% of the natural parasitism on *N. viridula* eggs [54].

More recently, a significant change in the composition of the pentatomid fauna in soybeans resulted in a reduction in *N. viridula* and an increase in *Euschistus heros* (Fabricius, 1798) (Hemiptera: Pentatomidae) in Brazil (Figure 1). Thus, *T. podisi* gained more importance as an egg parasitoid, being responsible for more than 80% of the parasitism recorded in *E. heros* eggs [55], as this is its preferred host. The high parasitism capacity of *T. podisi* on eggs of different stink bug species (Figure 2) led this species to be used in Brazil as a biocontrol agent in ABC of stink bugs in soybean crops [50]. In an agroecosystem

such as soybean, where different stink bug species can occur, CBC practices that preserve the diversity of the natural enemy communities are highly desirable to keep the stink bug complex under control [56].

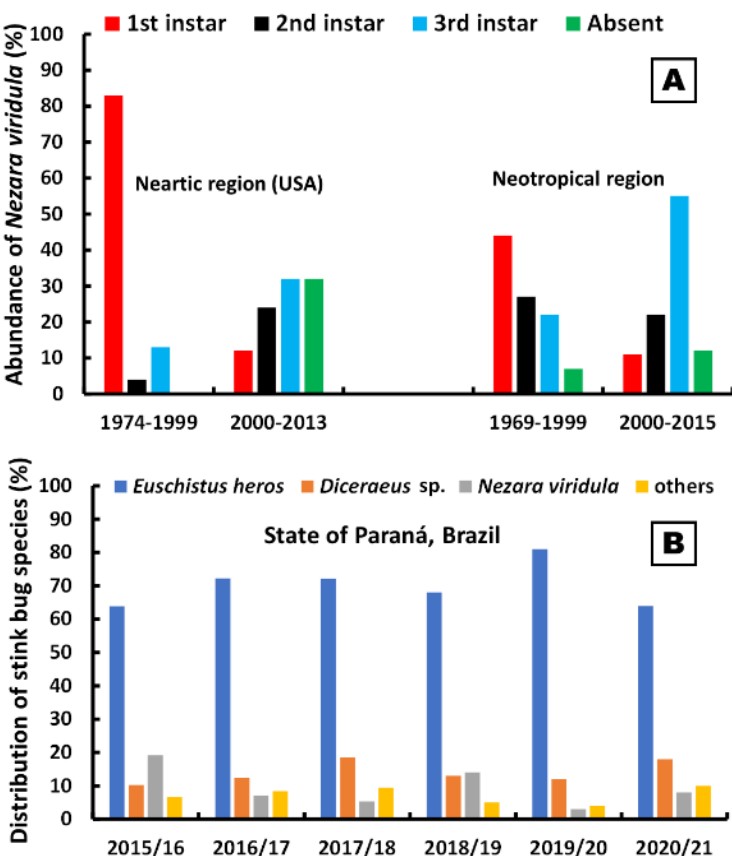

**Figure 1.** Evolution of the pentatomid fauna over time: (**A**) Abundance of *Nezara viridula* on different plants compared with other species of Pentatomidae; (**B**) Distribution of Pentatomidae species in soybean fields; Adapted from [53,57–62].

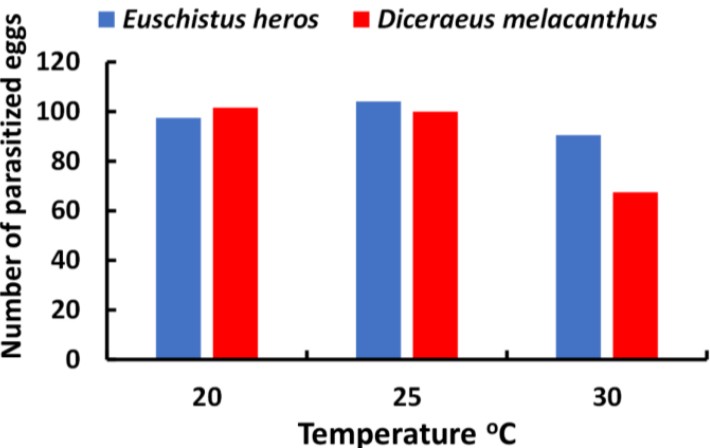

**Figure 2.** *Telenomus podisi* parasitism on stink bugs eggs. Adapted from [63,64].

Agricultural practices that preserve a richer community of biocontrol agents are important to establish stronger and more balanced agroecosystems, which are less susceptible to pest outbreaks [11]. Unfortunately, the loss of insect species to local and global extinction has been recorded at unprecedented rates [65], and agricultural systems often appear as

one of the main causes of this decreased biodiversity. Thus, this review focuses on the most promising alternatives to increase the presence of biocontrol agents of stink bugs in soybeans. Some studies show that agricultural systems may be suitable for the conservation of insect biodiversity when agroecosystems are properly managed [66]. Understanding how the adoption of sustainable IPM practices can help to increase or preserve biological control, an ecosystem service valued at USD 4.5 billion annually, is important to shape agriculture practices, especially for stink bugs in soybeans [1]. Biocontrol agents are particularly vulnerable [67,68] in large-scale crops such as soybean, which might receive a high number of insecticides every growing season if IPM is not properly adopted [1,2].

Despite the efficiency and importance of natural biocontrol agents, the logistic complexity of farm operations in large-scale crops such as soybeans can sometimes discourage farmers from integrating different pest management tools [2,69]. Of the wide range of impediments to the broad adoption of CBC practices among other sustainable IPM strategies in soybeans, the most important are usually related to economic considerations [2]. Concerns about the practicability, complexity, and costs of CBC or ABC practices caused by difficulties in their timing and implementation, as well as the lack of incentives, are reported as the major challenges to greater IPM adoption [70].

## 3. Recommendations of Soybean IPM to Preserve and Increase Biological Control in the Agroecosystem

Despite being the most challenging pest for managing soybeans, stink bugs can be more sustainably managed by adopting some simple soybean IPM recommendations [1]. Then, the use of chemical insecticides, especially the most harmful ones, can be avoided or at least mitigated, as discussed here. IPM recommendations must be both practical, profitable, and efficient to be widely adopted in large fields to mitigate challenges faced by farmers during their adoption [2]. The market demands that increasingly call for sustainably produced food products, together with the development of decarbonized agriculture [71,72], have renewed the interest in soybean IPM and its related sustainable pest control strategies, such as CBC and ABC, the reduction in the load of insecticides and use of selective products, refuge for natural enemies, host plant resistance, and timing of insecticide applications, among other sustainable practices and technologies [1].

(a)  Reduction in the number of pesticide applications during soybean growing seasons via the adoption of the economic threshold

Cropped plants can tolerate certain injury levels with no economically significant yield reduction [7]. Therefore, stink bug control should only be used when pest numbers are equal to or surpass two stink bugs per meter, which is the current economic threshold (ET), and considered the economically correct time to begin stink bug control [73]. In addition, when this ET is reached, the most selective control should be adopted to preserve the majority of biocontrol agents present in the agroecosystem [11].

Despite being less harmful than chemicals, even biological control products should be used with caution. For instance, the massive release of *Cotesia flavipes* (Cameron, 1891) (Hymenoptera: Braconidae) in sugarcane to control *Diatraea saccharalis* (Fabricius, 1794) (Lepidoptera: Pyralidae) as part of an ABC program carried out for years in Brazil probably displaced tachinid parasitoid flies from the environment where they used to be commonly found [74]. Such undesirable consequences can also happen in soybeans if a biocontrol agent of stink bug is released or applied abundantly in the fields without adopting a sound ET, especially considering the large area cultivated with this crop (around 45 million hectares in Brazil alone). However, ETs for stink bug control [7] were developed for the rational use of chemical insecticides, and a sound ET for using biocontrol agents still needs to be further researched [50]. Nevertheless, the simple adoption of the existing ET for stink bugs and the consequent reduction in insecticide use can significantly preserve biocontrol agents, providing ecological and financial benefits.

Since the late 1960s and early 1970s, soybean IPM has been adopted in Brazil. The results were recorded in several counties of the State of Paraná in a joint effort of the

Brazilian Agricultural Research Corporation—Soybean (Embrapa Soybean) and the State Government via the "Paraná Rural Development Institute (IDR-Paraná)" as a successful example of IPM as previously published in the literature for many different pests and insecticides [1]. In this review, the results related only to stink bugs will be discussed for the first time to demonstrate that even the most hard-to-kill pest can be sustainably managed via the adoption of CBC and ABC strategies in IPM. In this soybean IPM program, stink bugs and other pests were evaluated weekly by an IDR-Paraná extensionist throughout the soybean growing season, and control decisions were taken respecting ETs (Table 2) from IPM recommendations. At the end of the soybean season, fields were individually harvested and evaluated. Simultaneously, a survey was carried out using a questionnaire for farmers who were not assisted by the soybean IPM program (non-IPM). Afterward, a comparison was made between non-assisted (non-IPM) and assisted farmers (IPM) (Table 3) with a focus on the stink bug numbers.

**Table 2.** Economic Thresholds (ETs) adopted in the soybean IPM program in Paraná, Brazil.

| Pests | ET(s) | Reference |
|---|---|---|
| Defoliators (Lepidoptera, Coleoptera, and others) | (a) 30% defoliation (soybean in the vegetative stage) or (b) 15% defoliation (soybean in the reproductive stage) | [75] |
| Pod feeders (Lepidoptera, Coleoptera, and others) | 25% of injured pods | [76] |
| *Spodoptera* spp. | 10 caterpillars ($\geq$1.5 cm)/meter | [69] |
| *Helicoverpa* sp. and *Chloridea virescens* | (a) four caterpillars/meter (soybean in the vegetative stage) or (b) two caterpillars/meter (soybean in the reproductive stage) | [28] |
| Stink bugs | (a) two stink bugs ($\geq$0.5 cm)/meter (soybean for grain production) or (b) one stink bug ($\geq$0.5 cm)/meter (soybean for seed production) | [7] |

**Table 3.** Soybean IPM program [1] results (mean) in Paraná State, Southern Brazil [57–62].

| Variable | Comparison | 2015/16 | 2016/17 | 2017/18 | 2018/19 | 2019/20 | 2020/21 |
|---|---|---|---|---|---|---|---|
| Number of insecticide applications during the soybean growing season to control stink bugs | IPM | 1.40 (123 growers) | 1.30 (141 growers) | 1.06 (196 growers) | 1.15 (241 growers) | 1.13 (255 growers) | 1.27 (191 growers) |
| | Non-IPM | 1.90 (314 growers) | 1.9 (390 growers) | 2.0 (615 growers) | 2.10 (773 growers) | 1.94 (553 growers) | 2.22 (518 growers) |
| Days until the first insecticide application to control stink bugs | IPM | 70.4 days | 78.4 days | 86.4 days | 82.3 days | 82.1 days | 83.3 days |
| | Non-IPM | 66.0 days | 69.5 days | 65.8 days | 63.1 days | 68.3 days | 62.6 days |
| Stink bug control costs * (% of the total number of insecticide applications) | IPM | 80.0 kg ha$^{-1}$ (66.7%) | 89.7 kg ha$^{-1}$ (65.0%) | 59.8 kg ha$^{-1}$ (70.7%) | 85.2 kg ha$^{-1}$ (67.7%) | 74.0 (68.5%) | 44.3 kg ha$^{-1}$ (73.8%) |
| | Non-IPM | 120 kg ha$^{-1}$ (50.0%) | 126.3 kg ha$^{-1}$ (51.4%) | 115.5 kg ha$^{-1}$ (58.8%) | 151.9 kg ha$^{-1}$ (61.8%) | 115.6 kg ha$^{-1}$ (64.2%) | 78.1 kg ha$^{-1}$ (65.1%) |
| Yield (kg/ha) | IPM | 3426.0 | 3870.0 | 3702.0 | 3006.0 | 3864.0 | 3654.0 |
| | Non-IPM | 3282.0 | 3828.0 | 3624.0 | 2916.0 | 3804.0 | 3618.0 |

[1] Program where public consultants (from IDR—Paraná) sampled pests across the seasons and made all decisions on pest management in IPM areas for selected farmers. At the end of the season, the results of IPM areas were compared with other non-IPM areas of Paraná, Brazil. * Costs transformed into the equivalent value in soybean weight during the year of each soybean season to make the value of cost timeless.

The adoption of IPM and its ETs in soybeans resulted in an average reduction of 40.1% of insecticides applied to control stink bugs (considering the growing seasons from 2015/16 to 2020/21), ranging from 26.3% (2015/16) to 53.0% (2017/18). Not only is the

adoption of ETs for stink bugs important to accomplish this result but also the adoption of ETs for the other pests because the overall reduction in insecticides is important to better preserve biocontrol agents in the field. Moreover, IPM reduced the number of insecticide applications to control stink bugs and increased the time (days) from sowing to the application of the first insecticide to control stink bugs by 14.8 days. This period was also the average from 2015/16 to 2020/21 growing seasons (Table 3). Each day that chemical pesticides are avoided, biocontrol is better preserved, and working to control the stink bug population (Table 1). Thus, reducing applied chemicals is valuable for managing stink bug outbreaks in soybean fields, especially considering the usually low selectivity of chemical insecticides available on the market to be used against this group of pests [11].

Moreover, IPM fields had a 75 kg higher average yield than non-IPM fields (Table 3), indicating the association of more sustainable pest control with higher profits, an essential combination to support more farmers adopting the technology. Yields in IPM fields were between 36 kg ha$^{-1}$ and 144 kg ha$^{-1}$ higher in 2020/21 and 2015/16, respectively. Although IPM is not directly responsible for increasing yield, it did reduce losses caused by pests [1]. In addition, areas adopting IPM had lower stink bug control costs, which were calculated and transformed to their equivalent in value of kg soybean per ha for each crop season, ranging from 33.8 kg ha$^{-1}$ (2020/21) to 55.7 kg ha$^{-1}$ (2017/18) lower for IPM fields than non-IPM fields. Taken together, the increase in yield and the reduction in costs, adopting IPM for stink bug management in soybean resulted in a higher profit of 120.7 kg ha$^{-1}$ (45.7 kg ha$^{-1}$ of reduction in stink bug control costs + 75 kg ha$^{-1}$ of indirect increase in soybean yield) (Table 3). Therefore, adopting IPM in large-scale crops can be considered one of the most promising CBC strategies due to the combined results of reducing pesticide use (preserving biocontrol agents) and increasing farmers' profits in the field.

(b) Use of selective pesticides

One of the IPM goals is to reduce the negative impact of using chemical products on crops. Within this concept, it is important to maximize the effects of pesticides on pests, with the minimum impact on beneficial organisms, due to the great presence and importance of the latter in agroecosystems, as previously discussed in this review (Table 1). One of the ways to minimize the impact of chemicals on biocontrol agents is the use of selective products. Selectivity can be defined as the ability of a product to control the targeted pest while also causing the least possible impact on beneficial organisms such as predators, parasitoids, bees, and entomopathogens. Selectivity occurs due to physiological and ecological differences between organisms [77]. Physiological selectivity is intrinsic to the chemical product, which, when applied in a given agroecosystem, is more toxic to the pest than to its natural enemies in a situation where both are exposed to the compound or its residues [78]. As an example, larvae of the predator *Chrysoperla carnea* (Hagen, 1861) (Neuroptera: Chrysopidae) can tolerate pyrethroid insecticides due to the presence of esterase enzymes in their organism, which degrades chemical molecules making them less toxic to the predator [79]. Ecological selectivity is directly related to the characteristics of the beneficial organism and its habitat, and it occurs due to the differences in behavior between pests and natural enemies, causing the chemical compound to have contact only with certain species, especially the pests. To obtain this type of selectivity, extensive knowledge about the bioecological aspects of the pests and beneficial arthropods present in the soybean agroecosystem is important. The use of selective pesticides can reduce the impact of insecticides by considering when and where to apply the product as well as the periods of lower occurrence of predators and/or parasitoids [80].

It should be noted that the selectivity must always be comparative, which means that a product may be less toxic to the natural enemy compared to another compound. Selectivity does not imply the total absence of a negative impact on the population of the beneficial organism. Thus, a compound classified as selective only indicates that it is less harmful than other compounds. Determining the selectivity of chemical products for beneficial organisms in their different stages of development always aims to make biological and chemical methods compatible for use in IPM programs [77,79,81–83].

Despite the importance of biocontrol control, the use of chemical pesticides is still necessary within the current soybean system, at least in a short-term scenario [2]. In this context, the use of selective pesticides inside soybean IPM improves biological control preservation and effectiveness in the agroecosystem [2,69,84]. Thus, selective pesticides are valuable for stink bug management [11]. A significant advantage of these products is their effectiveness with minimal side effects for natural enemies of stink bugs [85]. Information obtained from research on the selectivity of pesticides for beneficial organisms is useful for the success of soybean IPM, and farmers should give priority to less toxic compounds whenever possible.

Studies about the impact of insecticides on egg parasitoids of stink bugs provide evidence that neurotoxic products do not show physiological selectivity for *T. podisi* and *T. basalis,* while growth-regulating insecticides (benzoylphenylurea) are classified as selective [86]. Although fungicides are considered harmless to beneficial organisms, compounds formulated with pyraclostrobin + metconazole, trifloxystrobin + cyproconazole, azoxystrobin + flutriafol, and trifloxystrobin + prothioconazole are not selective for *T. basalis*. Nevertheless, herbicides used on soybeans that are based on glufosinate ammonium salt and glyphosate isopropylamine salt are selective for *T. podisi* and *T. basalis* [87–89].

Seed treatment with systemic insecticides has recently been questioned due to their potential risks to non-target organisms. The effects of treating soybean seeds with chlorantraniliprole and thiamethoxam on the life history and walking behavior of the predator *Podisus nigrispinus* (Dallas, 1851) (Heteroptera: Pentatomidae) were evaluated. Thiamethoxam caused mortality of this predator, increased pre-oviposition period, and reduced oviposition period, female fecundity, and survival compared to chlorantraniliprole. In contrast, the lifespan of *P. nigrispinus* females was prolonged via chlorantraniliprole, which also increased the intrinsic growth rate (rm), increased the finite growth rate (λ), and reduced the population doubling time (DT) in comparison with thiamethoxam. The net reproduction rate (R0), the average generation time (T), the walking speed of *P. nigrispinus* females, and the distance covered were not negatively affected by any of the studied insecticides. Considering the observed lethal and sublethal effects, soybean seed treatments with chlorantraniliprol and thiamethoxan were considered low and moderate risk for the predator, respectively [79].

Selectivity studies must evaluate the effects of pesticides on the beneficial organism at all stages of its development (egg, larva, pupa, and adult) because sensitivity may vary among them. Endoparasitoids are generally less sensitive to the effects of chemical substances because they are protected within the host's body. A study observed that the immature stages of *T. podisi* were more tolerant to the action of chemical products (insecticides, fungicides, and adjuvants) than adults [90].

In Brazil, the use of bioinputs (organics) in soybean cultivation has increased in recent years. Products based on *Baculovirus anticarsia*, *Bacillus thuringiensis*, Azadirachtin-A, azadirachtin-B, nimbina and salamina, rotenoids, nitrogen, phosphorus and total organic carbon, sodium silicate, copper + calcium, and sulfur + quicklime exhibited low toxicity for pupae and adults of *T. podisi*. Therefore, these organic inputs can be used in soybean crops, as they do not compromise the potential of this parasitoid for controlling stink bugs [91].

Among the most commonly used and studied pesticides employed on soybeans, biopesticides are generally the most harmless biocontrol agents important for stink bugs, followed by fungicides, herbicides, and insect growth regulators (IGRs). Despite some harm caused by ethiprole, this insecticide is still more selective than pyrethroids, carbamates, or organophosphates (Table 4).

**Table 4.** Classification of insecticide selectivity to *Telenomus podisi* according to the "International Organization for Biological Control" (IOBC) on different days after spraying (DAS) or days after adult emergence (DAE).

| Treatment (g ha$^{-1}$) | Parasitoid Development Stage | | | | | Reference |
|---|---|---|---|---|---|---|
| | Pupae | | | Adult | | |
| | Sprayed Pupae | 1 DAE | 3 DAE | 1 DAS | 3 DAS | |
| Beta-cyfluthrin 7.5 | 1 | 1 | 1 | 1 | 1 | [86] |
| Beta-cyfluthrin 12.5 + imidacloprid 100 | 1 | 1 | 1 | 4 | 4 | [86] |
| Bifenthrin 5 | 1 | 1 | 1 | 4 | 4 | [86] |
| Chlorantraniliprole 10/15/20/30/50 | 1 | 1/1/1/1/1 | 1/1/1/1/2 | - | - | [63,86] |
| Chlorantraniliprole 7.5/10/20/30 + lambda-cyhalothrin 3.75/5/10/15 | 1/1/1/1 | 1/1/1/1 | 1/1/1/1 | 3/-/-/- | 3/-/-/- | [63,86] |
| Chlorfluazuron 37.5 | 1 | 1 | 1 | 1 | 1 | [86] |
| Chlorpyrifos 480/640/960 | 2-1-2 | 3-1-2 | 3-2-2 | 4-2 | 4-2 | [63,85,86] |
| Deltamethrin 7.5 | 1 | 1 | 1 | 2 | 2 | [86] |
| Ethiprole 100/33.3 | 1/1 | 1/1 | 1/1 | 1/1 | 1/1 | [85] |
| Flubendiamide 33.6 | 1 | 1 | 1 | 1 | 1 | [86] |
| Lambda-cyhalothrin 7.5 | 1 | 1 | 1 | 3 | 3 | [86] |
| Lufenuron 7.5 | 1 | 1 | 1 | 1 | 1 | [86] |
| Methoxyfenozide 21.6 | 1 | 1 | 1 | 1 | 1 | [86] |
| Novaluron 7.5 | 1 | 1 | 1 | 1 | 1 | [86] |
| Spinetoran 3 | 1 | 1 | 1 | 1 | 1 | [86] |
| Spinosad 24 | 1 | 1 | 1 | 1 | 1 | [86] |
| Sulphoxaflor 13.3/20 + lambda-cyhalothrin 20/30 | 3/3 | 1/2 | 2/2 | 3/3 | 3/4 | [85] |
| Tebufenozide 30 | 1 | 1 | 1 | 1 | 1 | [86] |
| Teflubenzuron 7.5 | 1 | 1 | 1 | 1 | 1 | [86] |
| Thiamethoxam 18.8/23.5/28.2 + lambda-cyhalothrin 14.1/17.7/21.2 | 2/2/1 | 1/1/1 | 1/1/1 | 3/4/4 | 4/4/4 | [85,86] |
| Triflumuron 14.4 | 1 | 1 | 1 | 1 | 1 | [86] |
| Zeta-cypermethrin 35 | 1 | 1 | 1 | 4 | 4 | [86] |

Classes: 1 = harmless; 2 = slightly harmful; 3 = moderately harmful; 4 = harmful; Work by [63] used 150 L of water/ha, ref. [86] 200 L of water/ha, and [85] 100 L of water/ha.- Not evaluated.

Although IGRs are regarded as less harmful to beneficial insects [92] compared with other chemical groups, negative side effects have also been reported, especially concerning their impact on immatures of different biocontrol species [93]. For example, although lufenuron does not kill adults, ingesting this insecticide significantly reduced the egg viability of *Chrysoperla externa* (Hagen, 1861) (Neuroptera: Chrysopidae). Therefore, it was classified as harmful (class 4) for this stage of development [94].

IGRs are used to control Lepidoptera in soybeans but not to control stink bugs. However, it is important to emphasize that when caterpillars need to be controlled in soybeans, the use of IGR is preferable to more harmful pesticides because it helps to protect important natural enemies of stink bugs. Among the recommended insecticides against stink bugs, biological inputs are the most selective products; among the chemical products,

ethiprole causes less harm to biocontrol agents compared with the ready-to-use mixture of pyrethroids and neonicotinoids [86].

Ethiprole is a new phenyl-pyrazole insecticide with a structure analog to fipronil. It has been widely used against stink bugs in soybeans and has high efficacy against a broad spectrum of sucking insects [95]. Overall, there are not many modes of action for insecticides used against stink bugs, and the incorrect and excessive use of these products without observing the rotation of their modes of action is the main cause of the worsening resistance-related issues of this pest group in soybean [2]. Therefore, ethiprole has been described as having several positive characteristics, such as a certain level of selective toxicity [96]. However, pesticide selectivity can largely differ between different beneficial organisms [77]. Ethiprole has been observed to cause developmental deficiencies, disordered immune action, abnormal reproduction, and neurobehavior in some other non-target organisms [97,98]. Sublethal doses of ethiprole were reported to have physiologically toxic effects on honeybee larvae and adult honeybees, impairing pupation and eclosion rates [99]. Because of this, it is always better to avoid using synthetic pesticides whenever possible and economically feasible, despite a certain level of selectiveness being reported.

The impact of the active ingredients trichlorfon, thiamethoxam, imidacloprid, and the mixtures of thiamethoxam + lambda-cyhalothrin, imidacloprid + beta-cyfluthrin, acetamiprid + cypermethrin, imidacloprid + carbaryl was evaluated under laboratory conditions on *T. podisi*. Higher mortality of *T. podisi* larvae was observed when exposed to trichlorfon, imidacloprid + carbaryl, and neonicotinoids/pyrethroids combinations. However, no treatment reduced *T. podisi* emergence under field conditions, indicating that applications of these insecticides at doses recommended for stink bug control do not affect *T. podisi* larvae [100].

In addition to physiological selectivity, ecological selectivity should be considered [11]. Ecological selectivity aims to reduce the exposure of natural enemies to the most harmful insecticides, which can be carried out through applications that consider the seasonality or circadian cycle of pests and natural enemies [101]. Another strategy can be using precision agriculture tools, with pesticides being applied only in areas where the pest presides, avoiding spaces in the crop where the ET has not been reached [102]. For stink bugs in soybeans in particular, mixing sodium chloride (NaCl) with insecticide products in the proportion of 0.5% ($v/v$) has an arrestant effect, prolonging feeding and thus prolonging the insect's exposure to insecticides. This mixture can be applied in stripes and, therefore, reduces the area covered with insecticide [103]. This is another important example of ecological selectivity use that can increase the preservation of natural biological control while still guaranteeing an acceptable insecticide performance.

(c) Host plant resistance

Host plant resistance represents the ability of certain soybean cultivars to produce larger yields than other soybean cultivars at the same level of pest infestation [104]. The classical three main categories of plant resistance—antibiosis, tolerance, and non-preference (later re-named antixenosis) were recently reduced to resistance (i.e., plant traits that limit injury to the plant subdivided into constitutive/inducible and direct/indirect subcategories), and tolerance (plant traits that reduce the amount of yield loss per unit injury) [105]. Soybean cultivars in this resistance category, mostly genetically engineered (GE) plants, constitute a foundational tactic of soybean IPM. The use of GE plants can minimize stink bug damage and, consequently, the number of insecticide applications during soybean crop [106].

The first attempt to adopt resistant soybean cultivars was in the 1990s [107]. However, the developed soybean cultivars did not succeed commercially because they lacked suitable agronomic characteristics and/or low productivity (especially seed yield) [108]. More recently, the soybean cultivar BRS 1003 IPRO was the first of several other commercial releases to have stink bug resistance, named block technology. Soybean plants were selected successively, exposing soybean lines to high levels of stink bug infestation for several years. Plants more tolerant to attack that showed high yield and good seed quality (low amount of

abnormal or damaged seeds) were selected to compose the plants with the block technology. Those cultivars tolerate more than two times the stink bug ET of two bugs/m with no significant reduction in yield compared to susceptible cultivars [109]. The adoption of soybean block cultivars has allowed higher stink bug damage without yield reduction compared to susceptible not-block cultivars. To avoid stink bug damage and keep the same yield, susceptible cultivars require twice the insecticide application compared to block cultivars (Table 5). Therefore, the adoption of resistance cultivars is a sustainable pest management tool that allows the reduction in insecticide use.

**Table 5.** Yield and seed quality of soybean cultivars susceptible and tolerant to stink bug damage in different Brazilian counties during the 2014/15 growing season [109]. Means followed by the same letter in the column do not differ by the Tukey test (p ≤ 0.05).

| County, State, Country | Cultivar | Number of Insecticide Applications | Yield (kg/ha) | Tetrazolio Test (%) | | | |
|---|---|---|---|---|---|---|---|
| | | | | Vigor | Viability | Damaged by Stink Bugs | Unviable Seeds |
| Andira, PR, Brazil | Block (BRS 391) | 2 | 4638 a | 78.9 a | 91.5 a | 65.3 b | 7.3 b |
| | Not Block (BRS 232) | 2 | 3222 b | 19.2 b | 58.9 b | 98.3 a | 37.5 a |
| Florínea, SP, Brazil | Block (BRS 391) | 1 | 5919 a | 86.4 a | 93.2 a | 20.5 b | 1.0 b |
| | Not Block (BRS 232) | 1 | 5281 b | 86.7 a | 94.5 a | 36.0 a | 3.4 a |
| Cândido Mota, SP, Brazil | Block (BRS 391) | 2 | 4485 a | 73.9 a | 90.7 a | 69.9 a | 7.5 a |
| | Not Block (BRS 232) | 4 | 4258 a | 73.7 a | 90.1 a | 63.7 a | 7.3 a |

Not only is host plant resistance important to stink bug management but also to Lepidoptera management. Overall, evidence of a regional reduction in insecticide use across areas was observed in Brazil with the widespread adoption of Cry1Ac soybean since 2013, with up to 50% reduction in the number of insecticide sprays, providing economic, social, and environmental benefits [110] as well as better preservation of biocontrol agents, including the ones that attack stink bugs.

(d) Increased plant diversification on soybean farms

Considering the agricultural systems characterized by the complete removal of plants after harvesting soybean as well as by the insecticide/herbicide applications, habitats with pollen, nectar, and honeydew should be provided during non-crop periods to keep the parasitoids and other biocontrol agents close to the area. One of those underexplored alternatives is the management of areas adjacent to the soybean fields. This can significantly improve the performance of mass-released parasitoids and maintenance or even increase the presence of naturally occurring biocontrol agents. Wild plants provide food sources and refuge against adverse conditions for natural enemies that occur naturally [78] and those that are released in the area. For example, different sizes of natural vegetation fragments and the distance between them influence the diversity of natural enemies and natural biological control of pentatomid eggs in soybean crops [111].

Brazil is one of the countries with the strictest legislation on nature preservation worldwide. Depending on the biome the farmers are cultivating, they are obligated by law to preserve between 80% (Amazon biome) and 20% (Pampa biome) of the land, known as a legal reserve [112]. This implies areas of preserved native vegetation interspersed with cultivated areas, which indirectly act as a repository of natural enemies. Instead of all the "legal reserve" being concentrated in a single plot, it helps natural enemies to better colonize the crop if the same size of the legal reserve is split into several plots within the

landscape. In addition to these legal reserves, simple maintenance of wild vegetation, especially flowering plants, on the periphery of crops, alongside roads, or along border fences contributes to the preservation of various natural enemies, as well as parasitoids released in large quantities to control specific pests [113].

The diversification of plant strata in windbreak barriers, which are widely used in different agricultural crops, is another example of a simple measure that could be easily adopted. Field windbreaks designed to work in CBC must include different strata of plants, varying in height and structure [114]. Thus, the presence of non-crop vegetation adjacent to the cultivated crops can provide benefits to biological control without major interference in area management. Despite such benefits, some topics still need to be better studied, including the identification of the best plant species to be used in surrounding areas and whether these landscape modifications could also increase the number of pests.

The use of flowering plants on the periphery of commodity crops to enhance the effectiveness of natural enemies by providing them with additional resources without negatively impacting crop management has been widely investigated [115] and implemented in some systems, including soybean [116]. In addition, the importance of integrating the use of different safer pest management tools in soybean fields has also been highlighted [1,2,70].

(e) Other examples of sustainable IPM strategies

The more sustainable IPM strategies integrated inside soybean IPM, the better. Association of CBC, ABC, ETs, reduced use of insecticides, use of selective insecticides, a refuge for natural enemies, host plant resistance, and timing of insecticides application, among other sustainable technologies, are important for the success of agriculture and should be researched and supported by public policies as clear strategy to reduce the use of insecticides. The use of GE insects, RNAi, plant oils and extracts, and genetically edited microorganisms are among the new technologies to be used in pest control. Evaluating their impact on non-target organisms is essential to select the most sustainable alternatives.

## 4. Final Considerations and Conclusions

Besides the more complex biological control strategies that could be adopted in soybean cultivation, adopting simple soybean IPM strategies with its consequent mitigation of insecticide use, using ETs to decide when insecticides should be applied, and prioritizing insecticides with less impact (such as the use of biologicals) has already shown important results in preserving natural biological control agents in soybean fields. The challenge to increase IPM adoption in soybeans may be overcome using economic incentives in the context of the recent global agenda of decarbonized agriculture [1]. Several governments worldwide have committed to reaching Net Zero greenhouse gas (GHG) emissions by 2050. With its potential to reduce 50% of insecticide use, soybean IPM will also help to reduce the GHG emissions associated with the production and application of unnecessary insecticides [1]. The increasing global demand for low-carbon soybean and the possibility of additional profits with carbon credits has pushed and renewed the interest in soybean IPM. This increase in soybean IPM adoption will, consequently, improve the preservation of natural biological control agents in this agroecosystem and make farmers more open to more sustainable control tools.

Intensified soybean IPM adoption should be incentivized in all possible ways because it is considered the best way to keep soybean production sustainable throughout the years. The use of biological control, including the association of CBC and ABC, as part of an IPM program contributes to the reduction in the use of pesticides. On the other hand, it is important to ensure the adoption of key practices at the field level, such as the use of selective active ingredients, the establishment of refuge areas for natural enemies, and host plant resistance, among others, to achieve sustainable soybean production. With a global population that is expected to reach 10 billion in 2050, the increasing demand for food will put more pressure on agriculture. Brazil has all the climate conditions of temperature, humidity, and light favorable to the crop. However, the intensification of soybean crop areas will also favor arthropod pests. Therefore, it is necessary to develop greener pest

management tools, such as new biological control products, RNAi, and other modern sustainable strategies that control pests with minimal impact on natural biological control. In Brazil, biological control programs with the use of bioinputs have been increasing in the last five years at a rate of 30% or more per year, mainly with applications of microorganisms (fungi, bacteria, and viruses). In soybeans, several species of natural enemies help regulate pest population growth, whether stink bugs or caterpillars. Therefore, studies about the compatibility of chemical and biological control methods as part of the management programs for these pests are of great importance. Both CBC and ABC strategies are expected to increase over the next few years due to the positive results observed by farmers at the field level, leading to increasing demand for those bioproducts and a growing market for bioproducts. However, adopting IPM practices in soybeans is crucial for the success of those sustainable pest control strategies.

**Funding:** This research received no external funding.

**Conflicts of Interest:** The authors declare no conflict of interest.

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
