# Peer review of "Biological Control as Part of the Soybean Integrated Pest Management (IPM): Potential and Challenges"

_agronomy, doi:10.3390/agronomy13102532_

Round 1

Reviewer 1 Report

Dear Authors,

The research subject is significant. Still, my main concerns are the high degree of similarity and increased topic overlap with several of your articles (especially references 2, 9, and 72). Therefore, compared to the reviews and articles published in this field, I could not find much innovation in this manuscript; and the other thing is that some data you have collected here are, in some cases, contradictory to the previous articles.

Moreover, you have written too repetitively throughout the manuscript. The manuscript has a lot of grammatical errors and must be improved by a native speaker.

 I have written other case comments in the text of the manuscript; to see them, please see the attached file.

 Good luck with your research

The manuscript has a lot of grammatical errors and must be improved by a native speaker.

Author Response

Considering that the time of both referees and editors is extremely precious and given on a voluntary basis and, therefore, aiming at saving time at the second round of evaluation, we highlighted the most substantial changes/considerations made in red in the new version of the manuscript.

Answer to Reviewer 1:

1) The research subject is significant. Still, my main concerns are the high degree of similarity and increased topic overlap with several of your articles (especially references 2, 9, and 72). Therefore, compared to the reviews and articles published in this field, I could not find much innovation in this manuscript; and the other thing is that some data you have collected here are, in some cases, contradictory to the previous articles.

Thanks for your appreciation and the suggestions received. The content was revised to avoid any overlap of topics and similarities; However, this manuscript is more focused on stink bugs with all the data related to this pest complex that is responsible for up to 60% of the insecticide use in soybean in South America. Of course, there is some degree of similarity related to the IPM principles that are similar indeed for all pests. However, the article highlights the sustainable management of stink bugs using data exclusive of this pest complex in the discussion. Data collected in this manuscript was carefully revised to avoid any contradiction and are precise as it means. Differences from previous articles are discussed and justified properly in the text.

2) Moreover, you have written too repetitively throughout the manuscript. The manuscript has a lot of grammatical errors and must be improved by a native speaker.

We reviewed the text to edit accordingly to remove any redundancy and improve readability as suggested. 

3) I have written other case comments in the text of the manuscript; to see them, please see the attached file.

All the edits highlighted in the pdf attachment of reviewer 1 were considered and included in the new version of the manuscript. Thanks for your appreciation and the suggestions made.

Reviewer 2 Report

Dear Editor,

The present manuscript studied the “Enabling Conservation Biological Control for the sustainable management of Stink Bugs in Soybean by applying IPM Practices: importance, perspectives and challenges” has clearly presented their review and compared different management tactics sustainable for management of Stink Bugs.

I would like to accept this manuscript in present form except a few minor suggestions highlighted in the attached pdf file.

Regards

Author Response

Considering that the time of both referees and editors is extremely precious and given on a voluntary basis and, therefore, aiming at saving time at the second round of evaluation, we highlighted the most substantial changes/considerations made in red in the new version of the manuscript.

Answers to Reviewer 2:

1) The present manuscript studied the “Enabling Conservation Biological Control for the sustainable management of Stink Bugs in Soybean by Applying IPM Practices: importance, perspectives and Challenges” has clearly presented their review and compared different management tactics sustainable for the management of Stink Bugs. I would like to accept this manuscript in its present form except for a few minor suggestions highlighted in the attached pdf file.

 All the edits highlighted in the pdf attached by reviewer 2 were carefully reviewed and included in the new version of the manuscript. Thanks for your appreciation and the suggestions made.

Reviewer 3 Report

The review manuscript by Bueno et al “Enabling Conservation Biological Control for the sustainable management of Stink Bugs in Soybean by applying IPM Practices: importance, perspectives and challenges” presents a review on various aspects aimed at promoting CBC for management of stink bugs in soybean. I recommend publication pending major revision.

The authors need to state clear objectives for this review and be guided by those to create a flow in paragraphs within each section. The title is about CBC – importance, perspectives and challenges but these are not clearly presented

Each section should start with a brief introduction and be presented in detail For example the section on “use of selective pesticides” should focus in detail on what selective use of pesticides is all about, types of selectivity, how this has been implemented or in progress for management of stink bugs in soybean with examples.

What is the purpose of presenting data in table 4 under the section “a) reduction of pesticides applied….”?

Include other CBC strategies apart from reduced use of pesticides and use of selective pesticides – refuge for natural enemies, host plant resistance, timing of insecticides application etc

The challenges of CBC have not been stated clearly – this should be in relation to natural enemies abundance, pest abundance, conducive habitat availability, agricultural practices, economics, expertise, scale of adoption, evidence of positive effect etc

The english needs minor editing, the paragraphs flow needs improvement.

Author Response

Considering that the time of both referees and editors is extremely precious and given on a voluntary basis and, therefore, aiming at saving time at the second round of evaluation, we highlighted the most substantial changes/considerations made in red in the new version of the manuscript.

Answers to Reviewer 3:

1) The authors need to state clear objectives for this review and be guided by those to create a flow in paragraphs within each section. The title is about CBC – importance, perspectives and challenges but these are not clearly presented.

The wording was edited to improve the text as suggested by the reviewer. Also, a new title was proposed to better describe the review content.

2) Each section should start with a brief introduction and be presented in detail for example the section on “use of selective pesticides” should focus in detail on what selective use of pesticides is all about, types of selectivity, how this has been implemented or in progress for management of stink bugs in soybean with examples.

The introduction was added as suggested by the reviewer.

3) What is the purpose of presenting data in table 4 under the section “a) reduction of pesticides applied….”?

Table 4 was changed in position inside the manuscript.

4) Include other CBC strategies apart from reduced use of pesticides and use of selective pesticides – refuge for natural enemies, host plant resistance, timing of insecticides application etc

The text was edited and the suggested information was added to the new version of the manuscript.

5) The challenges of CBC have not been stated clearly – this should be in relation to natural enemies’ abundance, pest abundance, conducive habitat availability, agricultural practices, economics, expertise, scale of adoption, evidence of positive effect etc

The text was edited in order to improve the quality of the manuscript as suggested by the reviewer.

6) The English needs minor editing, the paragraphs flow needs improvement.

The text was edited and English revised to improve the quality of the manuscript.

Reviewer 4 Report

This is a kind of review of conservative biological control strategies for controlling hemipteran pests in soybean. I do not quite agree with the approach the authors are taking, since the concept of CBC per se looks confusing and sometimes along the manuscript seems to switch to IPM, so it is not completely clear the differentiation between both concepts. In the introduction section, authors are defining CBC as a type of landscape architecture, but when they are proposing harmless pesticides as a CBC option, it does not look consistent, and seems that authors are meaning IPM almost all the time. So, in my opinion, this two concepts must be cleared up and differentiate in the manuscript. Also, the aim of this research should be clearly stated, and since this is a review the methods followed for bibliography search should be explained: which databases did authors use? Which are the search criteria authors used? I guess they made a systematic review of the published papers in that subject, because if not there might be a bias for some journals or regions. Also, in the title there is no reference to a particular country, but very often along the manuscript Brazil is mentioned, with very specific references to some results and some laws, so I think authors should mention that, if not in the title, at least in the abstract.

I have some other minor comments that I will explain next:

Line 46: since you have not introduced the stink bugs yet (and also I feel the sentence is kind of odd), I suggest to change it by something like “induce insect pest resistance”.

Lines 66-67: as I said before I do not quite agree with that definition for CBC. If the habitat sustains natural enemies then you are improving natural biological control, of course you are not disrupting natural biological control. I would rather suggest: “CBC is based on improving natural biological control by maintaining a habitat that can sustain natural enemies”.

Line 84: please locate the “(Hemiptera: Pentatomidae)” right after “Stink bugs” in line 81.

The place where Table 1 is mentioned is not right, since in the sentence you are talking about natural enemies, but in the table only parasitoids are included. On the other side, the table includes “parasitoids of importance”, but it remains unclear what are you meaning, because in the next paragraph you are mentioning 23 parasitoid species, and only egg parasitoids (in the table there are 18 parasitoid species). Which was the criterion for selecting those 18 species?

In Figure 1A, what does 1st, 2nd, 3rd and absent categories refer to?

Line 192: could you briefly described the IPM program?

Line 356: please remove the first “preserved”, it is repeated in the sentence.

Line 372: please substitute “lead to an increase the number” by “lead to increase the number”.

Author Response

Considering that the time of both referees and editors is extremely precious and given on a voluntary basis and, therefore, aiming at saving time at the second round of evaluation, we highlighted the most substantial changes/considerations made in red in the new version of the manuscript.

Answers to Reviewer 4:

1) This is a kind of review of conservative biological control strategies for controlling hemipteran pests in soybean. I do not quite agree with the approach the authors are taking, since the concept of CBC per se looks confusing and sometimes along the manuscript seems to switch to IPM, so it is not completely clear the differentiation between both concepts. In the introduction section, authors are defining CBC as a type of landscape architecture, but when they are proposing harmless pesticides as a CBC option, it does not look consistent, and seems that authors are meaning IPM almost all the time. So, in my opinion, this two concepts must be cleared up and differentiate in the manuscript.

Thanks for the suggestions received! The title was changed as a response to this.

2) Also, the aim of this research should be clearly stated, and since this is a review the methods followed for bibliography search should be explained: which databases did authors use? Which are the search criteria authors used? I guess they made a systematic review of the published papers in that subject, because if not there might be a bias for some journals or regions.

The text was edited to include the suggestion made by the reviewer.

3) Also, in the title there is no reference to a particular country, but very often along the manuscript Brazil is mentioned, with very specific references to some results and some laws, so I think authors should mention that, if not in the title, at least in the abstract.

Brazil was now mentioned in the abstract as suggested.

I have some other minor comments that I will explain next:

4) Line 46: since you have not introduced the stink bugs yet (and also I feel the sentence is kind of odd), I suggest to change it by something like “induce insect pest resistance”.

Done. The text was edited as suggested.

5) Lines 66-67: as I said before I do not quite agree with that definition for CBC. If the habitat sustains natural enemies then you are improving natural biological control, of course you are not disrupting natural biological control. I would rather suggest: “CBC is based on improving natural biological control by maintaining a habitat that can sustain natural enemies”.

The text was edited as suggested.

6) Line 84: please locate the “(Hemiptera: Pentatomidae)” right after “Stink bugs” in line 81.

Done. The text was edited as suggested.

7) The place where Table 1 is mentioned is not right, since in the sentence you are talking about natural enemies, but in the table only parasitoids are included.

The place of the first mention of Table 1 was changed as suggested.

8) On the other side, the table includes “parasitoids of importance”, but it remains unclear what are you meaning, because in the next paragraph you are mentioning 23 parasitoid species, and only egg parasitoids (in the table there are 18 parasitoid species). Which was the criterion for selecting those 18 species?

The title of Table 1 was changed to make it clearer.

9) In Figure 1A, what does 1st, 2nd, 3rd and absent categories refer to?

They refer to stink bugs instar. The text was edited to solve this mistake.

10) Line 192: could you briefly described the IPM program?

Done. The IPM program was briefly described as suggested.

11) Line 356: please remove the first “preserved”, it is repeated in the sentence.

Done. It was removed as required.

12) Line 372: please substitute “lead to an increase the number” by “lead to increase the number”.

Done. The text was edited accordingly to the suggestion.

Reviewer 5 Report

The review by Bueno et al. is an interesting work that highlights how stink bugs in soybean production systems can be sustainably managed.

In lines 83 and 84, the authors should indicate two or three of the most destructive or economically-important stink bug species in either the study region or USA and Brazil even though this has been stated in other sections of the review.

An estimate of the yield loss due to this pest group should be stated in line 85.

The authors should revisit the in-text citation style used in this review since both the numbering style and author, date/year style have been used in the paper. The authors should use only the numbering style (lines 309 & 327).

Although the selectivity of the insecticides to Telenomus podisi is presented in Table 4, the toxicity to the target pest is not indicated. A brief statement on the current efficacy/effectiveness of the insecticides on the pest group as a footnote would suffice.

Other comments have been indicated on the reviewed pdf.  

The English language is acceptable and just very minor corrections are needed.

Author Response

Considering that the time of both referees and editors is extremely precious and given on a voluntary basis and, therefore, aiming at saving time at the second round of evaluation, we highlighted the most substantial changes/considerations made in red in the new version of the manuscript.

Answer to Reviewer 5:

1) The review by Bueno et al. is an interesting work that highlights how stink bugs in soybean production systems can be sustainably managed.

Thanks for your appreciation and the suggestions made.

2) In lines 83 and 84, the authors should indicate two or three of the most destructive or economically-important stink bug species in either the study region or USA and Brazil even though this has been stated in other sections of the review.

The text was edited as suggested.

This group comprises at least 54 different species (Hemiptera: Pentatomidae) reported from different soybean-growing areas [24] among which stand out mainly the species of the genus Euschistus and Diceraeus [2].

3) An estimate of the yield loss due to this pest group should be stated in line 85.

An estimate of yield loss was provided.

Feeding directly from soybean pods, those insects can seriously reduce yields directly reducing around 15% of the yield [25] besides impairing the physiological and sanitary quality of the seeds when not properly managed [26].

4) The authors should revisit the in-text citation style used in this review since both the numbering style and author, date/year style have been used in the paper. The authors should use only the numbering style (lines 309 & 327).

Done. The text was edited to correct the improper citation format.

5) Although the selectivity of the insecticides to Telenomus podisi is presented in Table 4, the toxicity to the target pest is not indicated. A brief statement on the current efficacy/effectiveness of the insecticides on the pest group as a footnote would suffice.

The text was edited accordingly to the suggestion.

6) Other comments have been indicated on the reviewed pdf. 

The English language is acceptable and just very minor corrections are needed.

Thanks for your appreciation and the suggestions made. All the suggestions were carefully incorporated into the manuscript.

Reviewer 6 Report

Dear authors,

this review is well conceived and well written, and I suggest the journal to accept your manuscript. However, references to tachinids are completely missing. Furthermore, you could discuss also the management of the other pests for which you present the economic treshold, in order to make the manuscript more complete. I would suggest adding this two parts, and I also added two comments on the pdf file.

Best regards.

Author Response

Considering that the time of both referees and editors is extremely precious and given on a voluntary basis and, therefore, aiming at saving time at the second round of evaluation, we highlighted the most substantial changes/considerations made in red in the new version of the manuscript.

Answers to Reviewer 6:

1) this review is well conceived and well written, and I suggest the journal to accept your manuscript.

Thanks for your appreciation and the suggestions received.

2) However, references to tachinids are completely missing. Furthermore, you could discuss also the management of the other pests for which you present the economic threshold, in order to make the manuscript more complete. I would suggest adding these two parts, and I also added two comments on the pdf file.

Thanks for the suggestions received which were incorporated into the manuscript, reinforcing the information and discussions on Economic Thresholds.

Round 2

Reviewer 1 Report

Unfortunately, the Authors ignored almost all of my comments in the first edition.

As I commented in the first version, considering the title, the authors have collected relevant content. Still, my main concerns are the high degree of similarity and increased topic overlap with several of their articles (especially references 2, 9, and 72). Therefore, compared to the reviews and articles published in this field, I could not find much innovation in this manuscript; the other thing is that some data they have collected here are, in some cases, contradictory to the previous articles.

Moreover, they have written too repetitively throughout the manuscript. The manuscript has some grammatical errors and must be improved by a native speaker.

The authors need to give a suitable answer to my comments and mention the innovation of the manuscript compared to their previous similar articles.

 Extensive editing of the English language is required by a native English speaker.

Author Response

A great effort was made to correct all the points raised by both reviewers (reviewers 1 and 3) and a detailed explanation of the performed modifications is given below. Considering that the time of both referees and editor is extremely precious and given voluntarily and, therefore, aiming at saving time at the second round of evaluation, we highlighted the most substantial changes made due to reviewer 1 in green and reviewer 3 in blue in the new version of the manuscript.

Reviewer 1# Round 2

1) Comments and Suggestions for Authors:

Unfortunately, the Authors ignored almost all of my comments in the first edition.

We are very sorry about this perception, as it was not our intention. During the first round of evaluation, the PDF crashed and did not open. We tried to incorporate all the points raised by reviewers but due to the PDF problems we have faced, maybe some suggestions were not adequately addressed. We are sorry for that, but due to the short deadline to resend the reviewed manuscript during the first round of reviews, we tried to incorporate all the suggestions received. Hopefully, during this second round, we can satisfy all the reviewers as all the observations and recommendations received have been incorporated.

2) As I commented in the first version, considering the title, the authors have collected relevant content. Still, my main concerns are the high degree of similarity and increased topic overlap with several of their articles (especially references 2, 9, and 72). Therefore, compared to the reviews and articles published in this field, I could not find much innovation in this manuscript; the other thing is that some data they have collected here are, in some cases, contradictory to the previous articles.

Different from previously published papers, this one is focused on stink bugs and brings results to improve stink bug management. The text was edited to make it clearer. See for example:

“Soybean-IPM has been adopted in Brazil since the late 1960s and early 1970s and results were recorded in several counties of the State of Paraná, in a joint effort of the Brazilian Agricultural Research Corporation – Soybean (Embrapa Soybean) and the State Government through its official institute “Paraná Rural Development Institute (IDR-Paraná)” as a successful example of IPM as previously published in the literature for the gross number of pests and insecticide use [1]. Differently, we will separate and discuss here for the first time the results related only to stink bugs to state that even the most hard-to-kill pest can be sustainably managed through the adoption of CBC and ABC strategies in IPM. In this soybean-IPM program,...”

3) Moreover, they have written too repetitively throughout the manuscript. The manuscript has some grammatical errors and must be improved by a native speaker.

The text was edited and a native English speaker revised the manuscript to correct the mentioned problem.

4) The authors need to give a suitable answer to my comments and mention the innovation of the manuscript compared to their previous similar articles.

The authors made a great effort to give a suitable answer to your comments and mention the innovation of the manuscript compared to previous similar articles. Differently from the previously published paper, this one is focused particularly on the management of stink bugs, so this was the first one to discuss the results related only to stink bugs to state that even the most hard-to-kill pest can be sustainably managed throughout the adoption of sustainable pest management. We demonstrate the importance of both CBC as well as ABC going from simple adoption to ETs to resistant soybean cultivars and selective pesticides.

We reviewed the articles cited and as they deal with IPM, many things were perceived as similar. However, the previous articles mention IPM in general and focus on general pests, whereas this paper deals specifically with stink bugs.

5) Comments on the Quality of English Language: Extensive editing of the English language is required by a native English speaker.

Done! A Native English speaker revised and edited the content. 

6) Table 3. Stink bug control costs* (% of the total number of insecticide applications). Why costs? These values are not costs.

Done! The text was edited as suggested.

Reviewer 3 Report

The review manuscript by Bueno et al “Importance and challenges of Soybean-IPM to establish a sustainable management of stink bugs” presents a review on various aspects of stink bug management in soybean. I recommend rejection if authors fail to thoroughly address reviewer comments.

The authors need to state clear objectives for this review and be guided by those to create a flow in paragraphs within each section. There is no flow in all paragraphs because the objectives are not stated. The authors changed the title from CBC to IPM and this has caused more confusion as IPM and CBC are not the same yet they are both being used. IPM is way broader and this manuscript as it is does not present sufficient review on stink bugs IPM in soy bean. Let the authors chose either CBC or IPM and consistent focus on that and do thorough literature search. They also mention ABC strategies in some sections this should be clearly explained.

Each section should start with a brief introduction and be presented in detail For example the section on “use of selective pesticides” should focus in detail on what selective use of pesticides is all about, types of selectivity, how this has been implemented or in progress for management of stink bugs in soybean with case studies. This comment was in first round of reviewer comments and it has not been addressed instead the section on selective use of pesticides has paragraphs that do not connect, each paragraph seems to be on a different topic hence no flow

Include other CBC/IPM strategies apart from reduced use of pesticides and use of selective pesticides – refuge for natural enemies, host plant resistance, timing of insecticides application etc - The authors should discuss more strategies in case they chose IPM as reduced use of pesticides and selective use of pesticides are not sufficient.

The challenges of CBC/IPM have not been stated clearly – this should in detail touch on natural enemies abundance, pest abundance, conducive habitat availability, agricultural practices, economics, expertise, scale of adoption, evidence of positive effect etc

The conclusion needs to be improved as it does not offer a comprehensive summary of the review. The objectives need to be clear in order to make a proper conclusion.

Can be improved

Author Response

A great effort was made to correct all the points raised by both reviewers (reviewers 1 and 3) and a detailed explanation of the performed modifications is given below. Considering that the time of both referees and editor is extremely precious and given on a voluntary basis and, therefore, aiming at saving time at the second round of evaluation, we highlighted the most substantial changes made due to reviewer 1 in green and reviewer 3 in blue in the new version of the manuscript.

Reviewer - Second Round

1) The authors need to state clear objectives for this review and be guided by those to create a flow in paragraphs within each section. There is no flow in all paragraphs because the objectives are not stated. The authors changed the title from CBC to IPM and this has caused more confusion as IPM and CBC are not the same, yet they are both being used. IPM is way broader and this manuscript as it is does not present sufficient review on stink bugs IPM in soybean. Let the authors chose either CBC or IPM and consistent focus on that and do thorough literature search. They also mention ABC strategies in some sections this should be clearly explained.

The text is focused on the importance and challenges of biocontrol adoption (both ABC and CBC) along with other IPM strategies (adoption of ETs and prioritization of more selective insecticides) whenever possible. The main message is BC (both ABC and CBC) are essential to IPM success as well as the opposite. ABC should be used in areas with the adoption of IPM to improve considerably its chances to succeed. Then, in the previous round of evaluation, since other reviewers had mentioned that the manuscript would be more focused on IPM than BC, we proposed the previous change which clearly did not pleased this reviewer here. Then, we are proposing a new change that we believe still addresses what the other reviewers had mentioned in the previous round of evaluation as well as this review comment. The new proposed title is: “Importance and challenges of biocontrol adoption and preservation of natural enemies among Soybean-IPM strategies to successfully establish a sustainable management of stink bugs” Hopefully this new proposed title can clearly convey the topic of the manuscript.

We know that is difficult to address the suggestions of so many different reviewers, which sometimes present different suggestions but we did our best in a great effort to accomplish all the points raised by all reviewers.

Manuscript

2) Each section should start with a brief introduction and be presented in detail. For example the section on “use of selective pesticides” should focus in detail on what selective use of pesticides is all about, types of selectivity, how this has been implemented or in progress for management of stink bugs in soybean with case studies. This comment was in first round of reviewer comments and it has not been addressed instead the section on selective use of pesticides has paragraphs that do not connect, each paragraph seems to be on a different topic hence no flow

Thanks! This was done accordingly!

Include other CBC/IPM strategies apart from reduced use of pesticides and use of selective pesticides – refuge for natural enemies, host plant resistance, timing of insecticides application etc - The authors should discuss more strategies in case they chose IPM as reduced use of pesticides and selective use of pesticides are not sufficient.
Done! The text of introduction in each section was edited as suggested.

3) The challenges of CBC/IPM have not been stated clearly – this should in detail touch on natural enemies’ abundance, pest abundance, conducive habitat availability, agricultural practices, economics, expertise, scale of adoption, evidence of positive effect etc

The text was edited in a great effort to make it clear

4) The conclusion needs to be improved as it does not offer a comprehensive summary of the review. The objectives need to be clear in order to make a proper conclusion.

Done! The conclusions were revised and improved as indicated.